# Comparative Analysis of the Ecological Succession of Microbial Communities on Two Artificial Reef Materials

**DOI:** 10.3390/microorganisms9010120

**Published:** 2021-01-06

**Authors:** Zhansheng Guo, Lu Wang, Wei Cong, Zhaoyang Jiang, Zhenlin Liang

**Affiliations:** Marine College, Shandong University, Weihai 264209, China; guozhansheng@sdu.edu.cn (Z.G.); wanglu1997@mail.sdu.edu.cn (L.W.); congwei2016@sdu.edu.cn (W.C.)

**Keywords:** artificial reef, concrete, wood, microbial community, macrobenthos, ecological succession

## Abstract

Concrete and wood are commonly used to manufacture artificial reefs (ARs) worldwide for marine resource enhancement and habitat restoration. Although microbial biofilms play an important role in marine ecosystems, the microbial communities that colonize concrete and wooden ARs and their temporal succession have rarely been studied. In this study, the temporal succession of the microbial communities on concrete and wooden AR blocks and the driving factors were investigated. The composition of the microbial communities underwent successive shifts over time: among the six dominant phyla, the relative abundances of Proteobacteria, Cyanobacteria and Gracilibacteria significantly decreased in wood, as did that of Cyanobacteria in concrete. Operational taxonomic units (OTU) richness and Shannon index were significantly higher in concrete than in wood. Non-metric multidimensional scaling ordination placed the microbial communities in two distinct clusters corresponding to the two substrate materials. The macrobenthic compositions of concrete and wood were broadly similar and shifted over time, especially in the first five weeks. The Shannon index of the microbial communities in concrete and wood increased significantly with the organism coverage. The results provide fundamental data on microbial community succession during the initial deployment of ARs and contribute to understanding the ecological effects of ARs.

## 1. Introduction

Artificial reefs (ARs) are underwater structures that are either made of waste materials or are purpose-built constructions with the main function of protecting and enhancing marine resources and recovering and repairing habitats damaged by anthropogenic intervention or climate change [1,2,3,4]. With the development of engineering techniques for AR construction, various materials have been used to build ARs, such as concrete, wood, metal, rock, plastic, clay, and fiberglass. Concrete is the most common material for construction of ARs worldwide due to its good stability, easy formability into various structures and sizes, and high fixation rate of organisms. Wood has also been used in ARs for its low cost, lack of toxicity and good compatibility with marine environments, and it is commonly used together with concrete to increase its stability [2,5,6].

A major topic in the study of ARs is the ecological succession of periphyton and its structural and functional relationship with ARs of various materials [5,7,8,9,10,11]. The formation of biofouling is a complex process in which four main stages have been identified: (1) immediately after immersion, surface conditioning occurs with the formation of a primary film by absorption of organic/inorganic macromolecules (seconds to minutes); (2) initial attachment and growth of bacterial cells and unicellular eukaryotes to the substrate surface (minutes to hours); (3) settlement of invertebrate larvae and macroalga spores (days to weeks); and (4) colonization of the substrates with the development of a complex community of multicellular species (weeks to years) [12,13]. Studies of biofouling-colonized ARs have focused on investigation of macroalgae and invertebrate communities and have largely disregarded the microbial community structure and the corresponding succession. Only one study has investigated the microbial biofilm diversity and succession of ARs with terminal restriction fragment length polymorphism analysis of fungal internal transcribed spacer regions [14].

In marine ecosystems, microbial biofilms are major primary producers that play an important role in the biochemical cycle and induce the settlement and recruitment of macroalga spores and invertebrate larvae [12,15]. The formation, succession and development of microbial biofilm are easily influenced by physical (e.g., temperature, light, salinity, dissolved oxygen, substrate texture, and structure), chemical (e.g., chemical compounds), and biological factors (e.g., presence and abundance of particular bacterial or other organisms) [15]. The substrate material, roughness, colour, and orientation have been shown to influence biofilm adhesion and cause differences in the microbial community structure [12]. However, the microbial communities of biofilms on various substrates tend to become more similar over time, and the variability in the microbial community structure appears to depend more on environmental biochemical conditioning [16]. Muthukrishnan et al. (2019) found that fouling microbial communities on various substrates were not only substrate-specific but also location-specific [17]. The microbial communities and macrobiota have a close synergistic relationship, and several studies have revealed that the AR materials influence the species richness and abundance of epibenthic communities [10,11]. The presence, abundance and coverage of the macrobiota may influence the microbial community composition and diversity; however, little is known about the relationship between prokaryotes and eukaryotes in marine biofouling assemblages [15].

Concrete and wood, as common raw materials for the construction of ARs, differ considerably in material properties, surface roughness and organic composition. The effects of concrete and wood on the macrobenthic community have been studied in riverine [5] and inshore [11,18] environments, but studies of their effects on microbial colonization and succession have been relatively rare. In particular, the relationships between the microbial communities and macrobiota that colonize concrete and wooden structures have received little attention. The existing studies have focused primarily on antifouling coatings, engineering construction, and other aspects. For example, Chlayon et al. (2018) investigated the effects of biofilm and barnacles on the surface of concrete and found that both forms of biological colonization could improve concrete’s durability and reduce chloride diffusion rates [19]. Muthukrishnan et al. (2019) investigated the development of microbial fouling communities on four substrates and found that the total biomass was higher on wood and steel than on polyethylene terephthalate and polyethylene [17].

In the current study, the main objectives of the study were (1) to investigate the development and succession of microbial communities on concrete and wooden AR blocks over time; and (2) to initially explore whether these communities were associated with the substrate and macrobenthic community.

## 2. Materials and Methods

### 2.1. Study Area and Preparation and Deployment of AR Blocks

The study was conducted in Shuangdao Bay, on the northeast of the Shandong peninsula, China. ARs with a total volume exceeding 3 × 10^4^ m^3^ have been deployed here since 2006 and are mainly composed of concrete reefs and abandoned boats. A total of 58 abandoned boats with dimensions of 12 × 3 × 3 m (length × width × height) were placed on the muddy sand bottom in 2012, at depths ranging from 8.5 to 16.0 m. According to diving observations, only the frames of the boats remain. The beams of the boats are approximately 2 m above the sea floor and were used to hang the AR blocks in this study.

AR blocks were created with untreated wood (*Populus* L.) and concrete. Sixty pairs of blocks were built, with each pair containing one block of each substrate material, with dimensions of 20 × 20 × 5 cm. A hole (1 cm diameter) was drilled in the center of each wooden block and lifting eye bolts in stainless steel were anchored in the center of each concrete block. Each pair of wooden and concrete blocks was connected with a 1.5 m length of 1 cm nylon rope. The rope was tied to the eyebolts first, then a knot was made 0.5 m from each eyebolt, and another knot was also made after tying each wooden block through the hole (Figure 1A). The 60 pairs of AR blocks were hung apart from each other on the beams of the abandoned boats by a diver on 10 July 2019, making sure to prevent contact between different pairs (Figure 1B). The coordinates for the location are 37.489747° N and 121.953275° E, and the mean depth is 11.5 m.

### 2.2. Sampling Procedure and Analysis

After the deposition of the ARs, sampling was conducted weekly for the first 8 weeks and monthly thereafter, with three pairs of AR blocks sampled each time (three concrete and three wood blocks). Due to the influence of winter weather, continuous sampling could not be conducted after December, and we found that the blocks had all been carried away by the waves by the time of the diving survey was conducted the following spring. In addition, one concrete block was lost during the third sampling time. In total, 65 blocks were sampled. The samples were denoted Cn.1, Cn.2, and Cn.3 for concrete and Wn.1, Wn.2, and Wn.3 for wood (*n* = sample time, ranging from 1 to 11; 1, 2, and 3 represented three repetitions). All samples made of the same substrate material and collected at the same time were assigned to one group and named Cn or Wn. The wooden and concrete blocks were transferred immediately into sterile sampling bags after draining them of water, stored in an ice box and returned to the laboratory within 1 h. The water in the vicinity of the ARs was also sampled. All macrofauna species present on the blocks were recorded and photographed, and species were determined to the lowest taxonomic level according to the morphological characteristics. The microbial biofilms were thoroughly rubbed with sterile brushes, the blocks and brushes were rinsed with sterile seawater, the rinsing water samples were filtered through 0.22-μm polycarbonate membranes after passing through a 50-μm pore prefilter to remove macrobiota, and the membranes were stored at −80 °C until DNA extraction.

During sampling, the temperature (T) and dissolved oxygen (DO) content of the water in the vicinity of the ARs were measured in situ using a handheld multi-parameter water quality meter (Yosemitech, China), and the salinity (S) and pH were measured with a water quality instrument (YSI Pro Plus, USA). The total dissolved inorganic nitrogen (DIN) and phosphate (PO43-) were determined using a SAN++ flow injection analyzer (Skalar, Netherlands) in the laboratory.

### 2.3. DNA Extraction, PCR Amplification and High-Throughput Sequencing

Total DNA was extracted using the Fast DNA SPIN Kit (MP BIO, Irvine, CA, USA) following the manufacturer’s protocols, and the DNA concentration and purity were assessed by agarose gel electrophoresis. The V3-V4 regions of the bacterial 16S ribosomal DNA (16S rRNA) gene were amplified using the specific primers 341F (5′-CCT AYG GGR BGC ASC AG-3′) and 806R (5′-GGA CTA CNN GGG TAT CTA AT-3′). The volume and amplification protocol of polymerase chain reaction (PCR) were determined according to the method of Ammon et al. (2018) [15] with slight modifications. The PCR was carried out in 30 µL reactions with 15 µL of Phusion^®^ High-Fidelity PCR Master Mix (New England Biolabs Inc., Ipswich, MA, USA); 0.2 µM of forward and reverse primers, and about 10 ng template DNA. The PCR mixture consisted of initial denaturation at 98 °C for 1 min, followed by 30 cycles of denaturation at 98 °C for 10 s, annealing at 50 °C for 30 s, and elongation at 72 °C for 30 s, and finally by extension at 72 °C for 5 min. The PCR products were mixed and purified using a Gene JET Gel Extraction Kit (Thermo Scientific, Waltham, MA, USA). Sequencing libraries were generated with the Ion Plus Fragment Library Kit 48 rxns (Thermo Scientific, Waltham, MA, USA). The libraries were assessed and sequenced on a Qubit@ 2.0 Fluorometer (Thermo Scientific, Waltham, MA, USA) and an Ion S5 XL platform at Novogene Bioinformatics Technology Co Ltd. (Beijing, China), respectively. The acquired sequences were filtered for quality using the QIIME 2 pipeline [20], and chimeric sequences were detected and removed using the USEARCH tool based on the UCHIME algorithm [21] by comparison with the Gold database. The effective sequences were assigned to operational taxonomic units (OTUs) at a 97% similarity level using UPARSE (v. 11.0.667) [22]. The OTUs of less than two reads were discarded to avoid possible biases. The taxonomic information of the representative sequence for each OTU was annotated using the GreenGene database based on the RDP classifier (v. 2.13) [23] with an 80% confidence threshold. Concrete sample C9.2 contained too few valid sequences and was therefore excluded from the analysis. Illumina next-generation DNA sequences were deposited in the Sequencing Read Archive (SRA) of the National Centre for Biotechnology Information (NCBI) under BioProject accession PRJNA675392, SRA run accessions SRX9460362-SRX9460424.

### 2.4. Statistics and Bioinformatics Analyses

The α- and β-diversity analyses of the microbial communities were conducted in the R environment (v.4.0.2) using the vegan (v. 2.5-6) and psych (v. 2.0.9) packages. The α-diversity (including observed species, Shannon index, Chao1 index and Good’s coverage) of each sample was calculated with QIIME 2 [20]. Non-metric multidimensional scaling (NMDS) analysis based on Bray-Curtis distances was performed to assess the relationship between the microbial communities from the concrete and wooden AR blocks using the vegan package. The ANOSIM, Adonis and *t*-tests were carried out to determine the significant differences in microbial β-diversity between the concrete and wooden AR samples. To detect significant taxonomic differences between concrete and wood, a least discriminant analysis (LDA) was used to identify biomarkers (threshold > 4.0) [24]. The macrobenthic coverage of the AR blocks was visually estimated with ImageJ (v. 1.53e) [25]. The linear least-squares regression mode was adopted to qualify the relationship between macrobiotic coverage and microbial α-diversity (OTU richness and Shannon index) and to assess the regularity of the relative abundance of dominant phyla over time. The co-occurrence networks were constructed by calculating the Spearman’s rank correlations with the psych package based on the relative abundance of genera, retaining the correlations between genera, with |ρ| greater than 0.6 and a *p* value of less than 0.05, using Gephi (v. 0.9.2) [26]. A redundancy analysis (RDA) was carried out to evaluate the relationship between the environmental parameters and microbial community using the vegan package.

## 3. Results

### 3.1. Alpha and Beta Diversity of Microbial Communities of Concrete and Wooden Artificial Reefs

A total of 2,316,840 and 2,422,059 high-quality 16S rRNA sequences were normalised from the concrete and wooden AR samples, respectively. The Good’s coverage was 97.26 ± 0.66%, which indicated that the sequence libraries covered most of the microbial community in these samples. Among the OTUs identified, 5870 were shared between the concrete and wooden samples and 2034 were specific to concrete, accounting for 25.73% of the total concrete-associated OTUs, and 13.51% of the OTUs in the wooden samples were wood-specific (Figure 2A). The average number of OTUs and Shannon diversity values in each sample group ranged from 1526.67 to 3592.33 and 6.17 to 9.52, respectively (Appendix A), and both were significantly higher in the concrete than in the wooden samples (*p* < 0.01) (Figure 2B). Comparing the concrete sampling groups, the Shannon diversity values in C7 were significantly higher than in C1, C2, and C4 (*p* < 0.01), and that in C10 was also significantly higher than that in C1 (*p* < 0.05). For the wooden groups, the Shannon diversity values in W4 were significantly lower than in W3 (*p* < 0.05) and W7 (*p* < 0.01).

NMDS ordination was used to differentiate the microbial community structures between the concrete and wooden samples (Figure 2C) and showed that the samples in the same substrate grouped together, whereas no overlaps were detected between concrete and wood. ANOSIM and Adonis further confirmed that the microbial communities on the concrete ARs differed significantly from those on wood (ANOSIM: *R* = 0.462, *p* = 0.001; Adonis: *R*^2^ = 0.164, *p* = 0.001). According to NMDS analysis, the concrete or wooden samples collected at the same or adjacent sampling times had similar microbial community structures and tended to group together.

### 3.2. Temporal Succession of Microbial Communities

Two kingdoms were identified in this study, including bacteria and archaea (mainly Thaumarchaeota). To analyze the temporal succession of the composition of the microbial communities from the concrete and wooden AR samples, the relative abundances were studied at the phylum level. The most dominant phylum was Proteobacteria (63.03 ± 9.67% in concrete, 68.41 ± 8.33% in wood), followed by Bacteroidetes (15.19 ± 5.60% in concrete, 21.86 ± 8.79% in wood). The other major microbial phyla (with a relative abundance of greater than 1.0%) were Cyanobacteria, Actinobacteria, Firmicutes, and Verrucomicrobia in concrete, whereas only two, Actinobacteria and Firmicutes, were found in wood (Figure 3A). The microbial compositions in both concrete and wood varied over time, but they did not follow exactly the same trend. Linear least-squares regression was adopted to study the stability of the relative abundances of the six dominant phyla over time (Figure 3B). The relative abundances significantly decreased over time for Proteobacteria (Pearson’s *r* = −0.368, *p* = 0.035), Cyanobacteria (*r* = −0.679, *p* = 6.671 × 10^−6^), and Gracilibacteria (*r* = −0.389, *p* = 0.025) in wood, as did that of Cyanobacteria (*r* = −0.567, *p* = 8.781 × 10^−4^) in concrete. In contrast, the relative abundance of Firmicutes in concrete significantly increased over time, and the rest of phyla in concrete and wood remained more or less changed over time (*p* > 0.05).

To study the temporal and substrate impacts on the microbial communities of the concrete and wooden samples, LDA was used to analyze differences in taxon composition between the two sample materials. LDA identified 30, 33, and 17 biomarkers (LDA > 4.0, *p* < 0.05) within concrete, within wood and between concrete and wood, respectively (Figure 4), which revealed that the dominant species of the microbial communities varied significantly over time and substrate. In the concrete ARs, the biomarkers were concentrated in groups C1 (nine biomarkers) and C11 (eight biomarkers). Actinobacteria (Micrococcales), unidentified Cyanobacteria, unidentified Gracilibacteria, Alphaproteobacteria (Caulobacteraceae) and Gammaproteobacteria (Alteromonadales) were enriched in C1. Nitrososphaeria, Bacteroidia, Anaerolineae, Alphaproteobacteria (Kordiimonadales) and Deltaproteobacteria (Desulfuromonadales, and Geobacteraceae) showed high relative abundance in C11. In the wooden groups, the discriminative taxa were concentrated in W1 and W5. Gammaproteobacteria (Alteromonadales, Cellvibrionaceae, Oceanospirillales, and Vibrionales) and Epsilonproteobacteria (Campylobacterales) showed a significant presence in group W1, whereas Clostridia (Defluviitaleaceae), Alphaproteobacteria (Rhizobiales), and Betaproteobacteria (Burkholderiaceae and Methylophilaceae) were more abundant in W5. The discriminative taxa between the concrete and wooden samples were identified as follows: Acidimicrobiia, unidentified Cyanobacteria, Alphaproteobacteria (Rhizobiaceae), and Gammaproteobacteria were enriched in the concrete substrate, whereas Flavobacteriia (Flavobacteriales), Bacteroidia, and Alphaproteobacteria were more abundant in the wooden substrate.

### 3.3. Co-Occurrence Network Analysis

The microbial community co-occurrence networks based on the robust and significant correlations were constructed to explore synergetic relationships in the samples from the concrete and wooden ARs (Figure 5). In total, 99 edges from 68 nodes and 118 edges from 69 nodes were identified in the concrete and wooden samples, respectively. The network topological characteristics were calculated to describe the complex patterns of correlations between microbial genera. The values of those characteristics, namely, modularity, average degree, average network distance and density, were 0.69, 1.456, 1.764, and 0.22 in concrete and 0.594, 1.71, 1.87, and 0.025 in wood, respectively. These values indicated that microbial interaction may be more intensive in wood than concrete.

The co-occurrence (positive) and co-exclusion (negative) of patterns of microbial genera were distinct between concrete and wood. More negative interactions presented in the network of wood (23, 19.49%) than in concrete (10, 10.10%), suggesting greater co-exclusion between genera in wooden samples. The network nodes of concrete and wood at the phylum level were similar, wherein Proteobacteria and Bacteroidetes were the dominant phyla, accounting for 80.88% and 89.86% of the nodes in concrete and wood, respectively.

### 3.4. Succession of Macrobiotic Biofouling Community and Correlation with Microbial Community

A total of 14 macrofauna and algal taxa were identified on the concrete and wooden AR blocks, including Mollusca (6 taxa), Annelida (2 taxa), Arthropoda (2 taxa), Echinodermata (1 taxon), Urochordata (1 taxon), Bryozoa (1 taxon), and Rhodophyta (1 taxon) (Table 1). The macrobiotic compositions were broadly similar on the concrete and wooden blocks. However, the species composition of the macrobiotic biofouling communities that colonized the concrete and wooden blocks changed between sampling times, especially in the first five weeks. *Hydroides ezoensis* was the pioneer sessile organism observed on the AR blocks at the second week, followed by *Anomia chinensis* and *Chlamys farreri*. *Ciona intestinalis*, *Watersipora subovoidea* and *Crassostrea gigas* were found at the fifth week. Some species only presented at one sampling time across the sampling period, such as *Ceramiales* sp. (fifth week on wood), *Mytilus edulis* and *Ophiactis affinis* (fifth month on concrete). The dominant species also varied over time: *H. ezoensis*, *C. intestinalis,* and *W. subovoidea* were the dominant species in the first three weeks, fourth to fifth weeks and fifth week, respectively, whereas from the sixth week to the end of the study, *C. gigas* became the absolute dominant species.

The organism coverage was calculated to further reveal the succession of the macrobiotic biofouling community on the AR blocks (Figure 6A). The organism coverage of the concrete and wooden blocks both showed an upward trend in the first three months and fluctuated in the last two months. Notably, in the fifth month, the mean coverage on wood was just 0.062 ± 0.065%, indicating that the macrobiotic biofouling organisms had detached from the blocks, possibly due to the strong sea waves in winter. The remaining colonizers were mostly broken shells of dead *C. gigas*, and this was also the case on the concrete blocks. The organism coverage on the concrete blocks was higher than that on wood except in the fifth and seventh weeks.

Linear least-squares regression was performed between organism coverage and the α-diversity of the microbial communities to study the correlations between macrobenthos and microbes (Figure 6B). The values of the Shannon diversity index showed a significant positive correlation with organism coverage in the concrete (Pearson’s *r* = 0.409, *p* < 0.05) and wooden (*r* = 0.499, *p* < 0.05) groups. The number of OTUs of samples in the wood group showed a significant positive correlation with organism coverage, but that relationship was not significant in concrete.

### 3.5. Environmental Variables and Relationship with the Microbial Community

The sea water in the vicinity of the ARs was characterized in terms of six environmental variables (Table 2). The water temperature showed distinct seasonal variation, being higher in summer and early autumn and lower in winter. The values of DO ranged between 4.91 and 8.22 mg/L and had an inverse relationship with the water temperature. The values of pH and salinity remained relatively stable and varied only slightly. The values of DIN and PO_4_^3^-P ranged from 54.64 to 100.00 μg/L and 5.02 to 13.75 μg/L, respectively. The highest values of DIN and PO_4_-P were in the samples collected in the eighth week and third month, and the lowest values were found in the second week and fourth month, respectively.

An RDA was conducted based on the OTU level to explore the relationship between the microbial community structure and environmental variables (Figure 7). The variance inflation factor values of all environmental parameters were less than 20, and the Spearman test revealed that five of these variables—T, DO, pH, S and DIN—were significantly correlated in the sequence axis (*p* < 0.01). The two axes explained 50.11% of the variation in the data: DO, pH and DIN showed positive correlations with RDA1; T, DO and PO_4_-P showed positive correlations with RDA2; and the rest of the variables showed negative correlations with RDA1 and RDA2, respectively.

## 4. Discussion

Among the most important functions of artificial reefs are their ecological effects: periphyton, which includes microorganisms, algae, and invertebrates, gradually colonizes the bare reef surface after deployment and initiates the succession process [27]. These epibenthic communities can provide food for some reef-associated fishes or other nekton, attract and assemble fish and maintain water quality [2,11]. Microbial communities are the pioneer colonizers on ARs but are easily influenced by biological and environmental factors. Researching the succession of microbial communities therefore contributes to understanding the ecological effects of ARs [3].

The AR microbial community shows distinct temporal succession, especially when comparing seasons. The abundances of the microbial community settled on a hard substrate increase over time at the early stage of biofilm formation [28,29,30], although Abed et al. (2019) found a significant decrease in the α-diversity of the fouling bacterial communities during the 28 days of deploying acrylic panels [31]. In our study, the average number of OTUs and Shannon index values increased in the first three weeks and decreased in the fourth week both on the concrete and wooden blocks (Appendix A), perhaps resulting from DO reduction and shifts in the dominant species on the reef blocks. Holmström et al. (1992) found that the biofilm actually inhibited the larval attachment of *Balanus amphitrite* and *Ciona intestinalis* after the latter settle on the reef [32], which indicates that they may have a competitive relationship with the microbial community and cause a decline in the α-diversity. The mechanism must be confirmed in a future study. The wooden blocks in this study showed lower diversity in winter than in summer both for the OTU numbers and Shannon index, but this was not observed on the concrete blocks, nor in previous studies of free-living bacterial communities in Bohai Bay [33] and lotic freshwater [34]. In addition to temporal succession, the substrate material was also found to affect the diversity in previous studies [16,17]. Here, we observed that the average number of OTUs and Shannon diversity index of the microbial community on concrete were significantly higher than on wood.

Although their relative abundances shifted over time, Proteobacteria and Bacteroidetes remained the most dominant phyla colonizing the concrete and wooden blocks, which was in line with previous studies of biofilms [12,16,17,35,36]. Proteobacteria are known to be pioneer colonizers [37], and their relative abundances exceeded 56.00% on both materials at the early stage of deploying the ARs, being higher on wooden blocks. As an important component of the biofouling community, Cyanobacteria can produce extracellular polymeric substances to contribute to a more consolidative biofilm development [31,38]. Cyanobacteria was a dominant phylum on concrete in the first four weeks, after which its abundance tended to decline over time. Most of the cyanobacterial species were unidentified. We suppose that the Cyanobacteria had either colonized the concrete blocks before their deployment or underwent Cyanobacterial blooms during the study period. However, due to the inability to identify specific genera or species, the impact of Cyanobacteria on the microbial succession was hard to elucidate. Among the six dominant phyla, two and three phyla were detected to have significant variations over time on concrete and wood, respectively. The relative abundances of the two most dominant phyla had no significant change over time, except for Proteobacteria on wooden ARs.

A range of factors including physico-chemical variables, geographic location, substrate roughness, substrate colour and substrate orientation have been shown to influence the surface microbial community [15,36,39,40,41]. The substrate specificity of microbial communities on artificial materials remains under debate [12,42], although it is known that the physical and chemical properties of substrates influence the settlement and development of microorganisms. Wood and concrete differ considerably in substrate roughness, density, pH, ability to resist rot, and organic composition [7,43,44]. Our NMDS analysis indicated that the microbial communities showed structural differences between the concrete and wooden substrates, and the *t*-test confirmed that the difference was significant (*p* < 0.01) (Appendix A). We detected 2034 and 917 OTUs that were specific to the concrete and wooden blocks, respectively. The dominant microbial phyla on concrete and wood were similar, with the main community differences being related to rare phyla with a relative abundance of less than 1.0%. However, it remains unclear whether microbial communities are in general substrate-specific, as synergistic effects within and between microbial and eukaryotic communities are common and likely influence species niche preferences [15]. Concrete and wood differed with respect to the coverage of macrobiotic biofouling, which might also have influenced the composition of the microbial communities.

The compositions of the macrobenthic communities that colonized the concrete and wooden ARs were similar except for a few species with low abundance, such as *Ceramiales* sp., *M. edulis,* and *O. affinis*. For both substrates, the organism number and coverage tended to increase over time until the final sampling time. With the blocks having been hung approximately 10 m underwater, the diver observed that only a few Rhodophyta species were growing on the beam of the boat and no macroalgae were found on the sand bottom, whilst the spores are mainly released during spring. The above factors might have resulted in the low abundance of macroalgae in this study. Barnacles are often observed among the pioneer colonizing species on ARs [4,45], whereas H. ezoensis was the pioneering sessile organism to colonize the ARs in this study. The study site is near an oyster aquaculture area that produces C. gigas in early summer, which may explain why this was the most dominant species colonizing the AR blocks from the sixth week.

Micro- and macro-organisms undergo well-known synergistic interaction, and marine colonization processes can be described as successive steps, with the formation of microbial biofilms preceding the settlement of macro-organisms. Therefore, microbial biofilms have been suggested as key mediators for macrobiotic colonization [38]. Studies have increasingly focused on how microbial communities mediate larval settlement, including attractive and inhibitive influences [46,47,48,49,50]. In our study, the organism coverage was observed to significantly influence the microbial Shannon index of concrete and wood, but because the microbial community structure was also affected by dynamic environmental variables, such as the ocean current, substrate material and physico-chemical properties of seawater, it was difficult to elucidate the specific impact of the microbial community on the colonizing organisms. As the interactions between microbial biofilm and macrobiota are complex, and the studies to date were performed in the laboratory, more studies that combine interior and natural environments are required to explore the specific relationships between microbial community structure and macrobiotic settlement.

Concrete and wood both presented good biocompatibility in this study. However, the wooden blocks were more seriously damaged than the concrete in the later stage of the experiment. The substrate texture of wood is relatively soft and easily worn by waves and corroded by sea water and marine boring organisms, which makes it more suitable for installation in freshwater environments with slow currents, such as rivers, streams and lakes [5,51,52,53]. In contrast, concrete is a more suitable material for ARs due to its density, strength and durability (>30 years) [7,54], but has higher construction costs than other materials [52]. The addition of industrial waste (e.g., slag and fly ash), biogenic material (e.g., oyster shells) and seabed silt into AR concrete is also a potential strategy for environmental remediation [55].

## 5. Conclusions

The temporal succession of the microbial communities colonizing concrete and wooden ARs was studied. The microbial communities underwent successive shifts in composition, with Proteobacteria and Bacteroidetes as the most dominant phyla across the whole period. The microbial community structure on concrete was significantly different from that on wood, with 30, 33 and 17 biomarkers detected within concrete, within wood and between the two substrate groups, respectively. The macrobiotic biofouling communities attached on concrete and wood also presented temporal shifts, whereas no distinct differences of macrobiotic community structure were observed between the two materials. Environmental variables, i.e., temperature, dissolved oxygen, pH, salinity and dissolved inorganic nitrogen, drove the shift of the microbial community structure. Considering the complex interactions between micro- and macro-organisms, more influencing factors should be considered in future research.

## Figures and Tables

**Figure 1 microorganisms-09-00120-f001:**
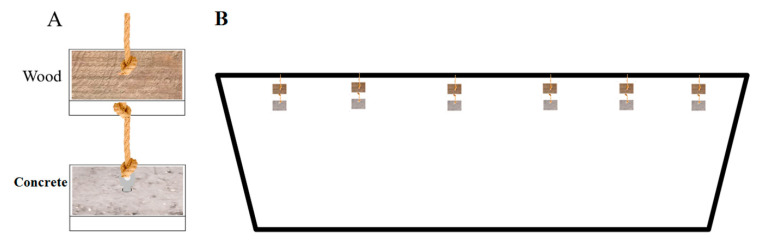
(**A**) Schematic diagram of a pair of concrete and wooden artificial reefs (AR) blocks; (**B**) vertical deployment of AR block array on the beam of a boat.

**Figure 2 microorganisms-09-00120-f002:**
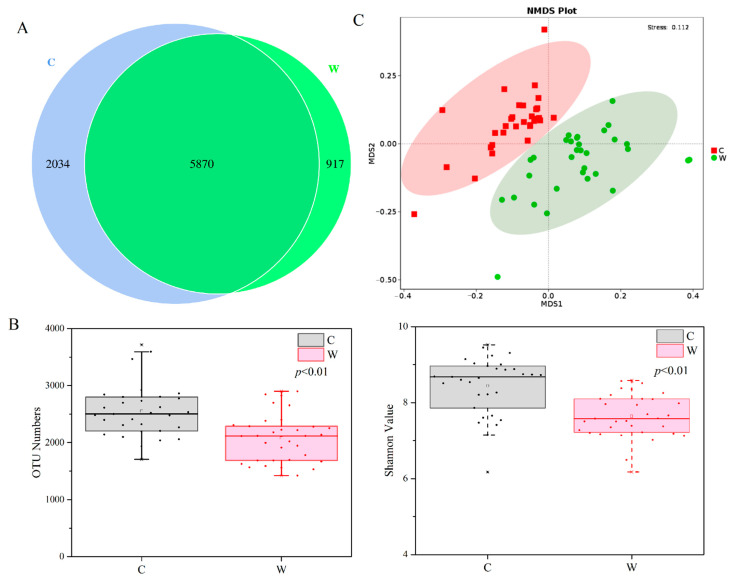
Venn diagram of microbial operational taxonomic units (OTUs) (**A**). Plots of α-diversity (OTU richness and Shannon index) between concrete and wood (**B**). Non-metric multidimensional scaling (NMDS) plots of microbial communities of concrete and wooden samples based on Bray-Curtis distances (**C**).

**Figure 3 microorganisms-09-00120-f003:**
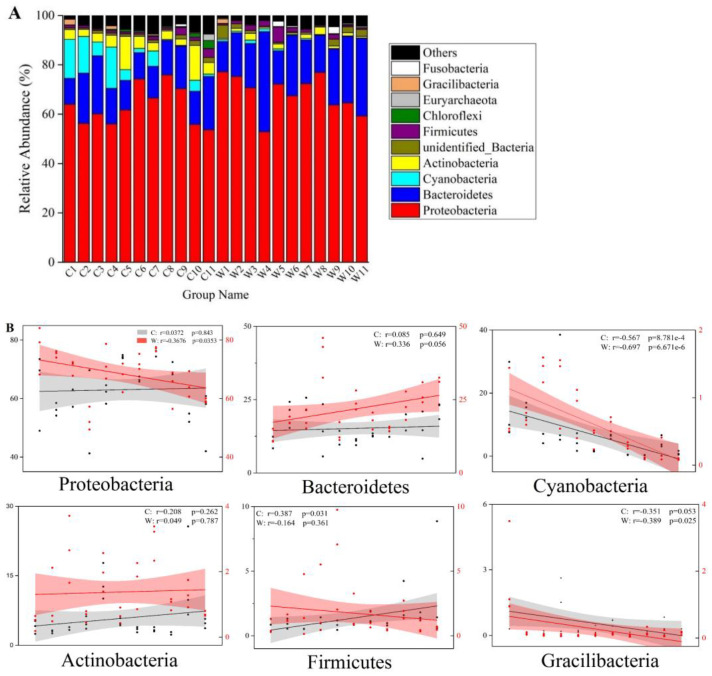
Temporal dynamics of the relative abundances of top 10 microbial phyla in concrete and wood (**A**). Pearson plots of six dominant microbial phyla over time (**B**). r represents the Pearson correlation coefficient.

**Figure 4 microorganisms-09-00120-f004:**
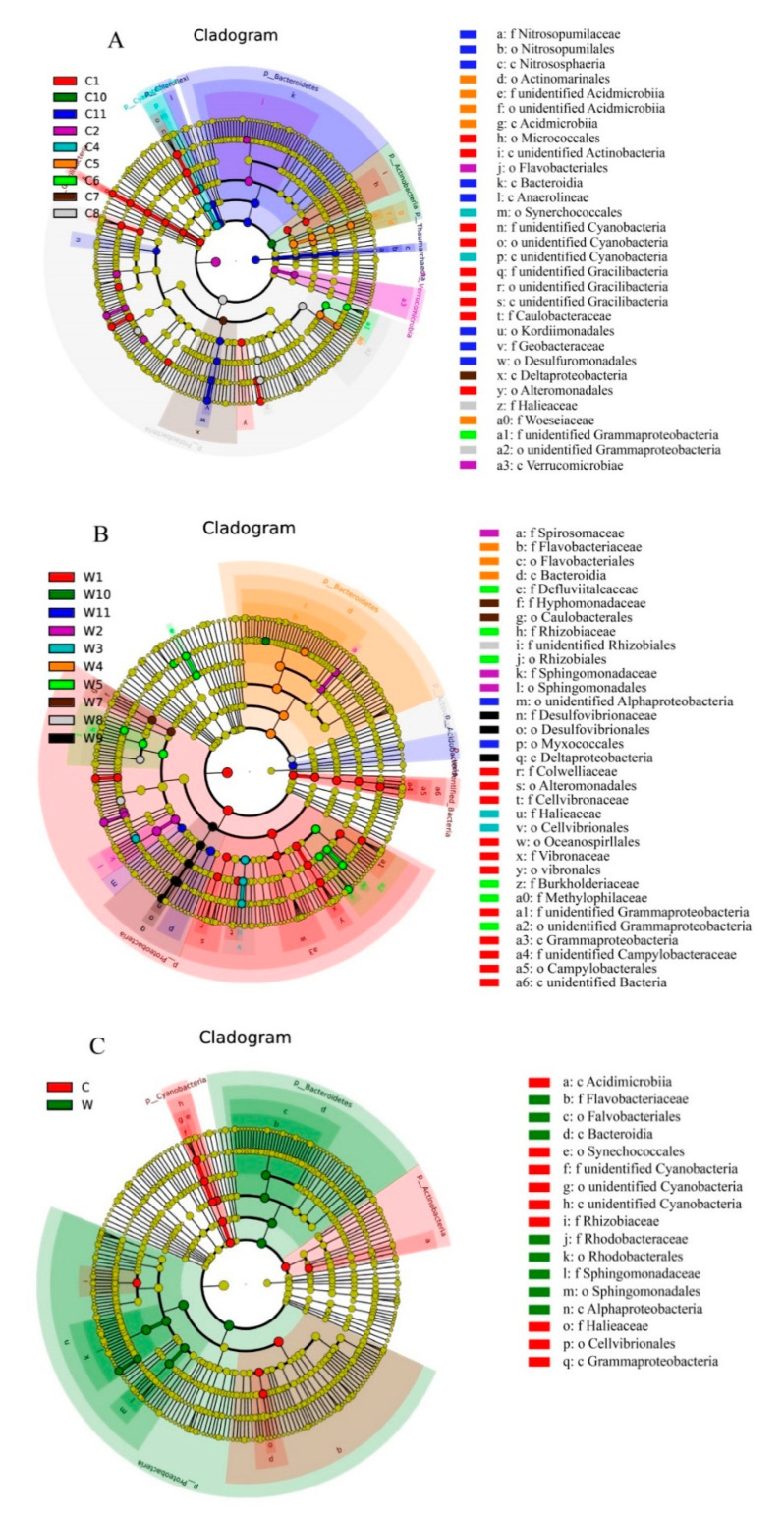
Taxonomic cladograms of the least discriminant analysis (LDA) effect size show significant microbial taxa that were discriminant within concrete group (**A**), within wooden group (**B**) and between concrete and wooden groups (**C**). Significantly discriminant taxon nodes are colored. Yellow nodes represent non-significant differences between groups. Diameter of each node represents the relative abundance of each taxon.

**Figure 5 microorganisms-09-00120-f005:**
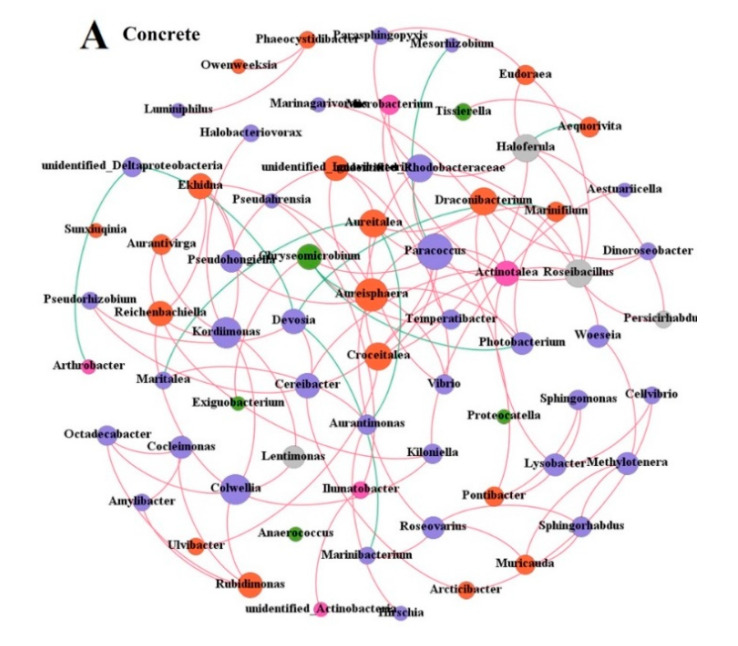
Co-occurrence network analysis at genus level on concrete (**A**) and wooden (**B**) blocks. The node colors represent different phyla. The sizes of genus nodes are proportional to their average relative abundance. The red and green edges represent positive and negative correlations, respectively.

**Figure 6 microorganisms-09-00120-f006:**
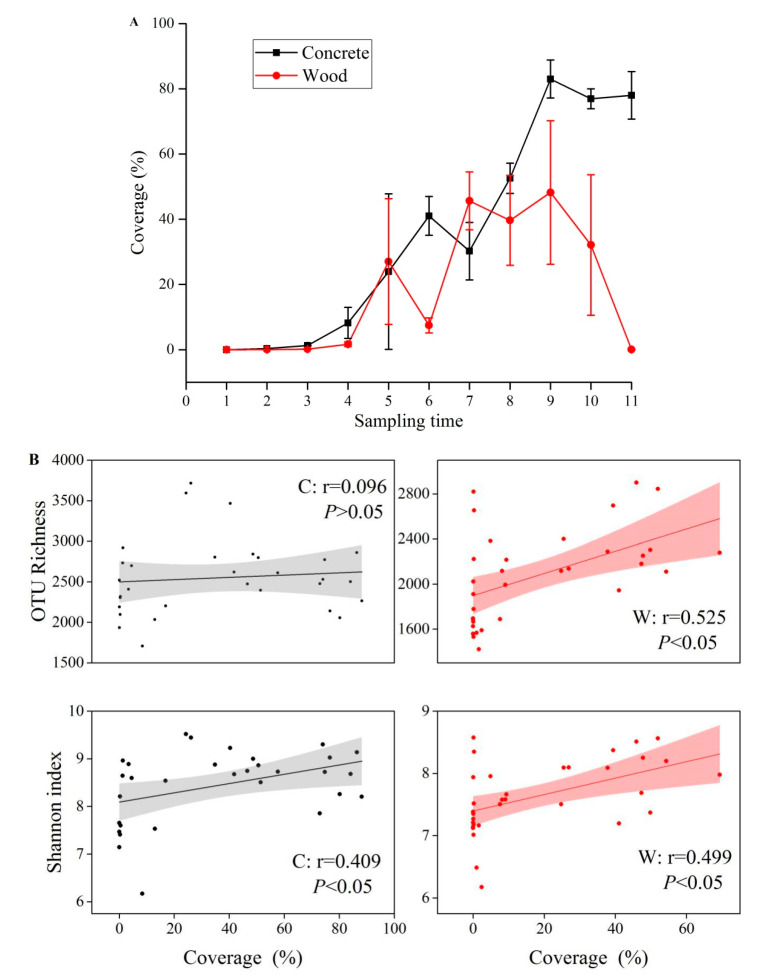
Dynamics of macrobenthic coverage over time (**A**) and Pearson plots of microbial OTU richness and Shannon index against macrobenthic coverage (**B**) on concrete and wood.

**Figure 7 microorganisms-09-00120-f007:**
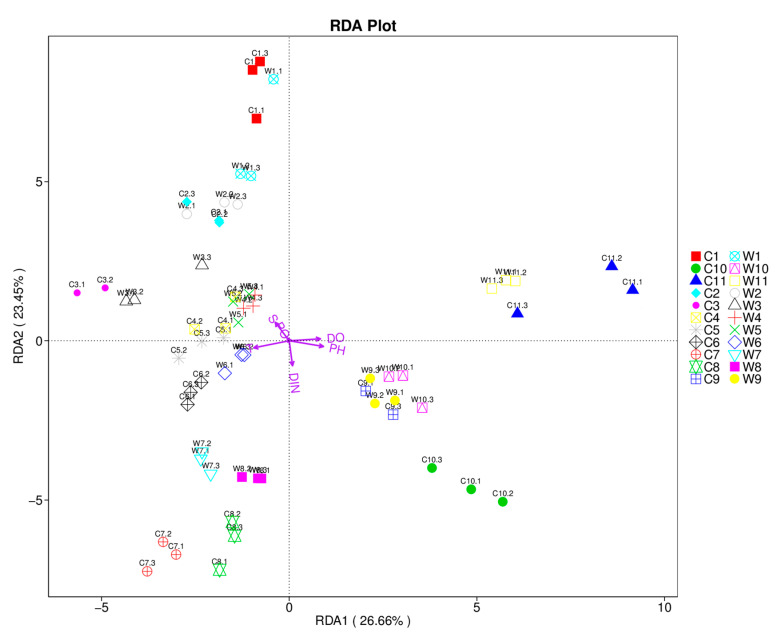
Redundancy analysis of environmental variables and microbial community.

**Table 1 microorganisms-09-00120-t001:** Presence of macrobenthos on the artificial reef blocks.

Scheme 1	1st Week		2nd Week		3rd Week		4th Week		5th Week		6th Week		7th Week		8th Week		3 Month		4 Month		5 Month	
	C1	W1	C2	W2	C3	W3	C4	W4	C5	W5	C6	W6	C7	W7	C8	W8	C9	W9	C10	W10	C11	W11
*Hydroides ezoensis*			●	●	●	●	○	○	○	○	○	○	○	○	○	○	○	○	○	○	○	
*Ciona intestinalis*							●	●	●	●	○	○		○							○	
*Watersipora subovoidea*							○	○	●	●	○	○	○	○	○	○	○	○	○	○	○	
*Crassostrea gigas*							○	○	○	○	●	●	●	●	●	●	●	●	●	●	●	○
*Anomia chinensis*					○						○	○	○	○	○		○	○	○		○	
*Mitrella bella*										○	○	○	○	○		○		○	○	○	○	
*Chlamys farreri*			○		○			○	○		○											
*Balanus amphitrite*															○	○	○	○	○		○	
*Leptochiton assimilis*															○							
*Gammarus* sp.				○								○							○			
*Lumbrineridae* sp.																	○				○	
*Ophiactis affinis*																					○	
*Mytilus edulis*																					○	
*Ceramiales* sp.										○												

C: concrete; W: wood; ○: present on the AR block; ●: dominant organism on the AR block.

**Table 2 microorganisms-09-00120-t002:** Environmental factors near the sampling site.

Sample	DO (mg/L)	T (°C)	PH	S (‰)	DIN (μg/L)	PO_4_-P (μg/L)
C1/W1	5.59	20.7	7.89	31.8	59.44	10.25
C2/W2	5.91	25.3	7.86	31.7	54.64	7.75
C3/W3	6.23	26.3	7.94	31.9	73.00	8.17
C4/W4	5.26	25.5	8.05	31.8	65.38	9.42
C5/W5	4.91	24.0	7.92	31.9	74.94	10.67
C6/W6	5.55	24.9	7.95	31.7	70.37	10.25
C7/W7	5.88	26.4	7.99	31.4	75.24	6.53
C8/W8	5.51	26.2	8.07	31.4	100.00	11.50
C9/W9	6.46	21.7	8.13	30.8	71.48	13.75
C10/W10	6.45	17.4	8.17	31.0	72.73	5.02
C11/W11	8.22	7.80	8.52	31.7	74.99	8.79

## Data Availability

MDPI Research Data Policies.

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
