# Peer review of "Comparative Analysis of the Ecological Succession of Microbial Communities on Two Artificial Reef Materials"

_microorganisms, 2021, doi:10.3390/microorganisms9010120_

Round 1
Reviewer 1 Report
I enjoy very much reading your manuscript. Introduction was informative with clear objectives declared. AR descriptions and deployment were well described, as well as DNA sequencing, statistics and bioinformatic were clearly and appropriately referenced.
I do have a couple of comments and questions I would like you to consider:
On line 137, you state "slight modifications". Usually, it is important to know exactly which modifications were included on a previously reported methodology (in your case, Ammon et al. 2018); then, I urge you to be explicit and precise on that or those slight modifications you applied.
Any comment on why detection of cyanobacteria was lower in AR-w than in AR-C (depth, nutrients or other factors)? What would be the reasons for not be able to identify cyanobacterial genera and species found on the AR? Was not matching of sequences found at public data banks? Would AR deployment depth (11 m) be a key factor influencing your results? If so, would you recommend a depth gradient in future studies? Which AR would you recommend to be deploy in larger number at the study site or is it to early to do so?
I am very interested in knowing your answers. Good job. Congratulations.
Author Response
1. On line 137, you state "slight modifications". Usually, it is important to know exactly which modifications were included on a previously reported methodology (in your case, Ammon et al. 2018); then, I urge you to be explicit and precise on that or those slight modifications you applied.
Reply: Thanks very much for your comments. We have added the details about the amplification volume and protocol of PCR according to the method of Ammon et al. (2018) in the revised manuscript, which were “The PCR was carried out in 30 µL reactions with 15 µL of Phusion® High-Fidelity PCR Master Mix (New England Biolabs Inc., Ipswich, MA); 0.2 µM of forward and reverse primers, and about 10 ng template DNA. The PCR mixture consisted of initial denaturation at 98℃ for 1 min, followed by 30 cycles of denaturation at 98℃for 10 s, annealing at 50℃for 30 s, and elongation at 72℃ for 30 s, and finally by extension at 72℃for 5 min.”
2. (1) Any comment on why detection of cyanobacteria was lower in AR-w than in AR-C (depth, nutrients or other factors)? (2) What would be the reasons for not be able to identify cyanobacterial genera and species found on the AR? Was not matching of sequences found at public data banks?
Reply: Thanks very much for your comments. (1)The relative abundance of Cyanobacteria on concrete ARs was higher than wooden ARs, which might be following reasons: â‘ the concrete ARs were stored outside the room, while the wooden ARs were stored indoor before deployment, and Cyanobacteria has been colonized on concrete before deployment; â‘¡ the surface of concrete ARs was strongly alkaline in the initial period, and Cyanobacteria detected in this study might be suitable for living in alkaline environment; â‘¢ other factors including nutrients, deployment depth, roughness etc. might also contribute to the results. (2) Most of the cyanobacterial genera or species were unidentified, which might be short of reference sequences in the GreenGene database.
3. Would AR deployment depth (11 m) be a key factor influencing your results? If so, would you recommend a depth gradient in future studies?
Reply: Thanks very much for your comments. In addition to the material of artificial reef, the deployment depth might be another key factor influencing the microbial community, as macrobenthic community and marine environmental factors (e.g. seawater transparency, sea flow, oxygen, etc.) exist differences in different marine depth, which may influence the microbial community structure, so we plan to study the microbial community colonized on ARs which were deployed in different underwater depth in next step.
4. Which AR would you recommend to be deploy in larger number at the study site or is it to early to do so?
Reply: In my opinion, concrete artificial reef is more suitable for deploying in larger number at the study site due to its good stability, easy formability into various structures and sizes, and high fixation rate of organisms.
Reviewer 2 Report
The authors of this manuscript (microorganisms-1035447) investigated the temporal succession of microbial biofilms that colonize both concrete and wooden artificial reefs. Overall, with the exception of some small grammatical comments and a query on statistical analysis, this manuscript is a well written and sound study.
Line 19: OTU is misspelt as OUT
Line 50-51: text is in a different font/size
Line 57-59: Citations are required for this statement
Line 149-151: text is in a different font/size
Line 178: Unsure what “OTU numbers” means, is it the average number of OTUs?
Line 179: “Shannon values” should be “Shannon diversity indices” or “Shannon diversity values”
Line 182: “Shannon values” should be “Shannon diversity indices” or “Shannon diversity values”
Line 183: “Shannon values” should be “Shannon diversity indices” or “Shannon diversity values”
Line 205: Is this Linear least-squares regression with repeated measures or non-repeated measures? It is currently unclear and I would suggest that unless it is repeated measures then the data should be retested.
Line 236: Delete extra period.
Figure 4: Key is misaligned, black bar is present on Fig 4. b
Line 292: Unsure what “OTU numbers” means, is it the average number of OTUs?
Line 334: Unsure what “OTU number” means, is it the average number of OTUs?
Line 344: Unsure what “OTU numbers” means, is it the average number of OTUs?
Line 371-374: text is in a different font/size
Line 380: text is in a different font/size
Author Response
- Line 19: OTU is misspelt as OUT
Reply: We are sorry for misspelling the abbreviation of operational taxonomic unit, the word “OUT” has been changed to “OTU”.
- Line 50-51: text is in a different font/size
Reply: Thank you for your careful work. We have rechecked and corrected the font/size of the text in line 50-51.
- Line 57-59: Citations are required for this statement
Reply: We have added the citation for the statement in line 57-59 according to the Reviewer’s requirement.
- Line 149-151: text is in a different font/size
Reply: Thank you for your careful work. We have rechecked and corrected the font/size of the text in the revised manuscript.
- Line 178: Unsure what “OTU numbers” means, is it the average number of OTUs?
Reply: Yes, the OTU numbers means the average number of OTUs, in order to avoid confusing the readers, the statements of “The average OTU numbers” have been corrected as “The average number of OTUs” in the revised manuscript.
- Line 179: “Shannon values” should be “Shannon diversity indices” or “Shannon diversity values”
Reply: Thank you very much. “Shannon values” have been changed to “Shannon diversity values” in line 179.
- Line 182: “Shannon values” should be “Shannon diversity indices” or “Shannon diversity values”
Reply: Thank you very much. “Shannon values” have been changed to “Shannon diversity values” in line 182.
- Line 183: “Shannon values” should be “Shannon diversity indices” or “Shannon diversity values”
Reply: Thank you very much. “Shannon values” have been changed to “Shannon diversity values” in line 183.
- Line 205: Is this Linear least-squares regression with repeated measures or non-repeated measures? It is currently unclear and I would suggest that unless it is repeated measures then the data should be retested.
Reply: Yes, the Linear least-squares regression was in repeated measures in the present study.
- Line 236: Delete extra period.
Reply: Thank you very much for your careful work. We have deleted the extra period.
- Figure 4: Key is misaligned, black bar is present on Fig 4. B
Reply: Thank you very much. We have redrew the Fig. 4, the details could be seen in the revised manuscript.
- Line 292: Unsure what “OTU numbers” means, is it the average number of OTUs?
Reply: “OTU numbers” means the number of OTUs of each sample, and not the average number of OTUs.
- Line 334: Unsure what “OTU number” means, is it the average number of OTUs?
Reply: Yes, “OTU number” means the average number of OTUs of each sampling time.
- Line 344: Unsure what “OTU numbers” means, is it the average number of OTUs?
Reply: Yes, “OTU number” means the average number of OTUs of microbial community on each material.
- Line 371-374: text is in a different font/size
Reply: Thank you for your careful work. We have rechecked and corrected the font/size of the text in the revised manuscript.
- Line 380: text is in a different font/size
Reply: Thank you for your careful work. We have rechecked and corrected the font/size of the text in the revised manuscript.